# PyramidDrop: Accelerating Your Large Vision-Language Models via Pyramid Visual Redundancy Reduction

## Abstract

In large vision-language models (LVLMs), images serve as inputs that carry a wealth of information. As the idiom "A picture is worth a thousand words" implies, representing a single image in current LVLMs can require hundreds or even thousands of tokens. This results in significant computational costs, which grow quadratically as input image resolution increases, thereby severely impacting the efficiency of both training and inference. Previous approaches have attempted to reduce the number of image tokens either before or within the early layers of LVLMs. However, these strategies inevitably result in the loss of crucial image information, ultimately diminishing model performance. To address this challenge, we conduct an empirical study revealing that all visual tokens are necessary for LVLMs in the shallow layers, and token redundancy progressively increases in the deeper layers of the model. To this end, we propose PyramidDrop, a visual redundancy reduction strategy for LVLMs to boost their efficiency in both training and inference with neglectable performance loss. Specifically, we partition the LVLM into several stages and drop part of the image tokens at the end of each stage with a pre-defined ratio, creating pyramid-like visual tokens across model layers. The dropping is based on a lightweight similarity calculation with a negligible time overhead. Extensive experiments demonstrate that PyramidDrop can achieve a 40% training time and 55% inference FLOPs acceleration of LLaVA-NeXT with comparable performance. Besides, the PyramidDrop could also serve as a plug-and-play strategy for inference acceleration without training, with better performance and lower inference cost than counterparts. We hope that the insights and approach introduced by PyramidDrop will inspire future research to further investigate the role of image tokens in LVLMs and explore additional methods to enhance their efficiency.

## 1 Introduction

In recent years, Large Vision-Language Models (LVLMs) have emerged as a central focus in deep learning research(Liu et al., 2024c; Dai et al., 2023; Bai et al., 2023; Zhang et al., 2024a; Chen et al., 2023a). We have witnessed remarkable progress across various application domains, including image and video understanding(OpenAI, 2024; Gemini Team, 2023). The rapid development of MLLMs is gradually paving the way for artificial intelligence to integrate into daily life(Li et al., 2023c; Zhu et al., 2023a; Zhang et al., 2023; Liu et al., 2024e).

However, despite the advancements in large vision-language models (LVLMs), a significant challenge lies in the escalating computational costs. Images, as continuous and information-rich signals, exhibit substantial spatial redundancy but are difficult to compress losslessly. It results in excessive image tokens and a steep increase in training and inference costs, which becomes particularly pronounced with higher image resolutions (Zhang et al., 2024a; Wang et al., 2024; Hu et al., 2024). The number of image tokens increases quadratically with the resolution, driving the sequence length into the tens of thousands(Li et al., 2023a). Given that the computational complexity of transformers scales with sequence length, the associated computational costs become prohibitively high(Liu et al., 2024a; Xu et al., 2024). Consequently, there is a pressing need to reduce the redundancy in visual information for more efficient LVLMs.

Previous exploration of image token compression could be roughly categorized into two ideas: compressing the token number before fed into the LVLM(Shang et al., 2024; Arif et al., 2024; Li et al., 2023d; Yao et al., 2024) or dropping part of the tokens at the very shallow layer of the LVLM(Chen et al., 2024a). However, both ideas inevitably hurt the performance of LVLMs: the former suffers from the information loss introduced by their compression, and the latter drops part of the information before the LVLMs fully understand them.

To break through the limitations of the aforementioned ideas, we explore the nature of LVLMs in understanding images from an intuitive question: *are all image tokens necessary for all LVLM layers?* We conduct an empirical study by removing different ratios of image tokens at different layers of the LVLM at inference time and observing the benchmark performance change. As shown in Figure 1, the LVLMs are sensitive toward token dropping on shallow layers, regardless of the dropping ratio. However, in deeper layers, image tokens gradually become less critical to the final results. The results indicate that the LVLMs understand the image layer-by-layer and the redundancy within image tokens increases correspondingly. We further visualize the attention between the instructions and the image tokens, and we observed a consistent phenomenon that in shallow layers, the LVLMs pay attention to most image tokens to understand the image globally. With the layer increasing, it tends to focus on the few tokens that are related to the instruction and the rest are unnecessary.

Based on the observation, we introduce PyramidDrop, a simple yet effective image token reduction strategy for LVLMs to accelerate both training and inference without performance loss. Pyramid-Drop divides the LVLM into several stages, dropping a portion of the image tokens at the end of each stage according to a predefined ratio. We employ a lightweight attention module to rank the image tokens, which incurs negligible overhead. With this design, we retain all image tokens in the shallow layers to avoid information loss, while progressively reducing the number of tokens as the layers deepen to maximize training and inference efficiency.

Extensive experiments verify the effectiveness and efficiency of our PyramidDrop. For example, LLaVA-NeXT-7B (Liu et al., 2024b) trained with PyramidDrop could reduce training time by 40% without sacrificing performance across 15 Vision-Language tasks. Moreover, PyramidDrop enables the LLaVA-NeXT model to be trained with doubled input resolution with only 269 GPU hours, which is 70% of the vanilla LLaVA-NeXT, and reaches a better performance on high-resolution benchmarks like DocVQA (Mathew et al., 2021) and InfoVQA (Mathew et al., 2022). Furthermore, PyramidDrop can function as a plug-and-play strategy for inference acceleration, offering enhanced model performance and fewer FLOPs than FastV (Chen et al., 2024a).

## 2 RELATED WORK

**Token Reduction** The large language model (LLM) realm has made several efforts in applying token reduction for inference acceleration and KV cache compression(Han et al., 2023). Stream-LLM(Xiao et al., 2023) only keeps attention sinks and the most recent tokens to reduce the size of the KV cache. FastGen(Ge et al., 2023) introduces an adaptive KV cache management approach that optimizes memory usage by adjusting retention strategies according to the specific properties of attention heads. Heavy-Hitter Oracle (H2O)(Zhang et al., 2024b) employs a strategy that selectively prunes key-value pairs (KVs) during generation, utilizing a scoring mechanism driven by cumulative attention to inform the removal process. ScissorHands(Liu et al., 2024d) concentrates on identifying and retaining important tokens that show a consistent pattern of attention weight across previous token windows during generation. These works attempt to address the redundancy of text tokens during the inference process in LLMs. As for visual tokens, existing works (Liang et al., 2022; Kong et al., 2022; Cao et al., 2023; Shi et al., 2024; Xiong et al., 2024) make explorations on Vision Language Models (VLMs) before the era of large vision-language models, focusing on token reduction for vision transformers (ViTs). A recent work, FastV (Chen et al., 2024a), makes an early attempt at visual token reduction in LVLMs, which drops visual tokens at the second layer of LVLMs during inference. In contrast, our work makes a more comprehensive study of the visual redundancy in LVLMs and proposes a pyramid visual token reduction solution for both training and inference of LVLMs.

**Large Vision Language Models** Enabled by the open-sourcing of large language models like LLaMA(Touvron et al., 2023) and Vicuna(Chiang et al., 2023), LVLMs(Chen et al., 2023b) have ad-

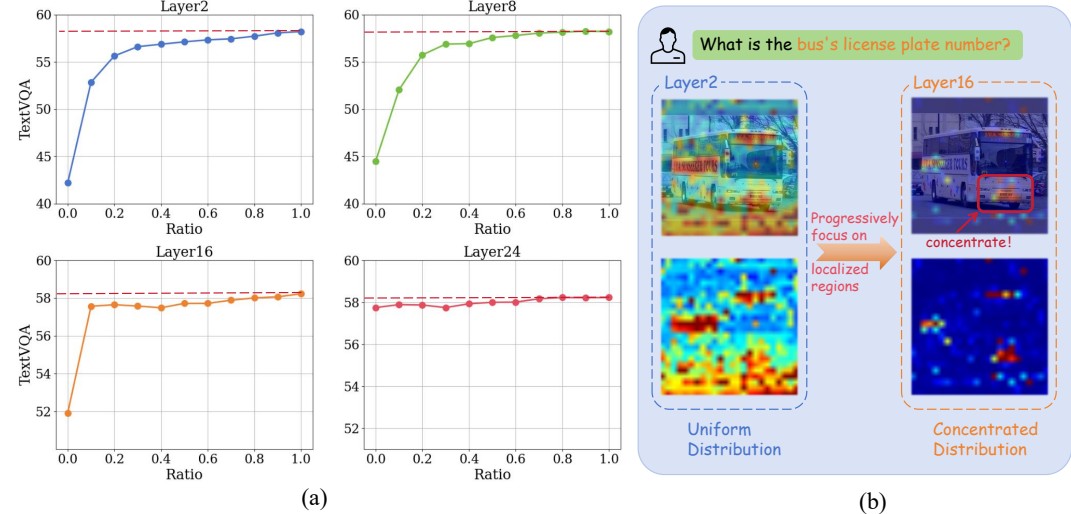

Figure 1: Observatioins about visual redundancy acoross layers. Left: TextVQA performance of LLaVA-1.5 with varying ratio of retained image tokens at different layer. The preserved image tokens are those that receive the highest attention from the text tokens. Right: Visualization of attention map in shallow and deep layers.

vanced the ability to understand and generate diverse content by seamlessly integrating information across multiple modalities, such as text, images, and audio. Models like LLaVA(Liu et al., 2024c), InstructBLIP(Dai et al., 2023), and MiniGPT-4(Zhu et al., 2023b) have pushed the boundaries of this field, enabling users to interact with these intelligent systems through multimodal prompts, including images and text. Recent advances (Zhang et al., 2024a; Wang et al., 2024; Hu et al., 2024) have significantly increased the number of image tokens for high-resolution image understanding, resulting in substantial costs for training and inference in LVLMs. This underscores the critical importance of developing more efficient training and inference methods for LVLMs.

## 3 METHOD

### 3.1 STUDY OF VISUAL TOKEN REDUNDANCY IN LVLMs

The fundamental design of PyramidDrop stems from an intuitive question: are all image tokens necessary for all LVLM layers? To explore it and reveal the nature of LVLMs, we conduct a two-variable experiment by removing different ratios of image tokens at different layers of the LVLM at inference time and observing the benchmark performance change.

In detail, we select LLaVA-v1.5-7B (Liu et al., 2024c) as the base model, and employ a popular LVLM benchmark, TextVQA (Singh et al., 2019), as the evaluation data. TextVQA consists of a substantial number of images that contain fine-grained information like text. The questions in TextVQA focus on the textual elements within images, requiring LVLMs to capture the global image information while mining the great detailed visual clues. This characteristic increases the model's sensitivity to image token compression, enabling a more precise evaluation of redundancy.

Considering LLaVA-v1.5-7B consists of 32 layers, we drop varying proportions of image tokens during inference at layer 2, 8, 16, and 24 to assess redundancy at different layers. The ranking of tokens is based on the attention values of text tokens towards image tokens, with the retained image tokens corresponding to those with the highest attention values. As illustrated in Figure 1(a), at layer 2, the LVLMs are sensitive toward token dropping on shallow layers, regardless of the dropping ratio. This indicates most of the image tokens in shallow layers play a important role in providing information for answering the instruction. With the layer increases, the redundancy of image tokens increases rapidly. At layer 16, even preserving only 10% of image tokens will not cause an obvious performance decline. Notably, at layer 24, the model performance is nearly irrelevant to the image

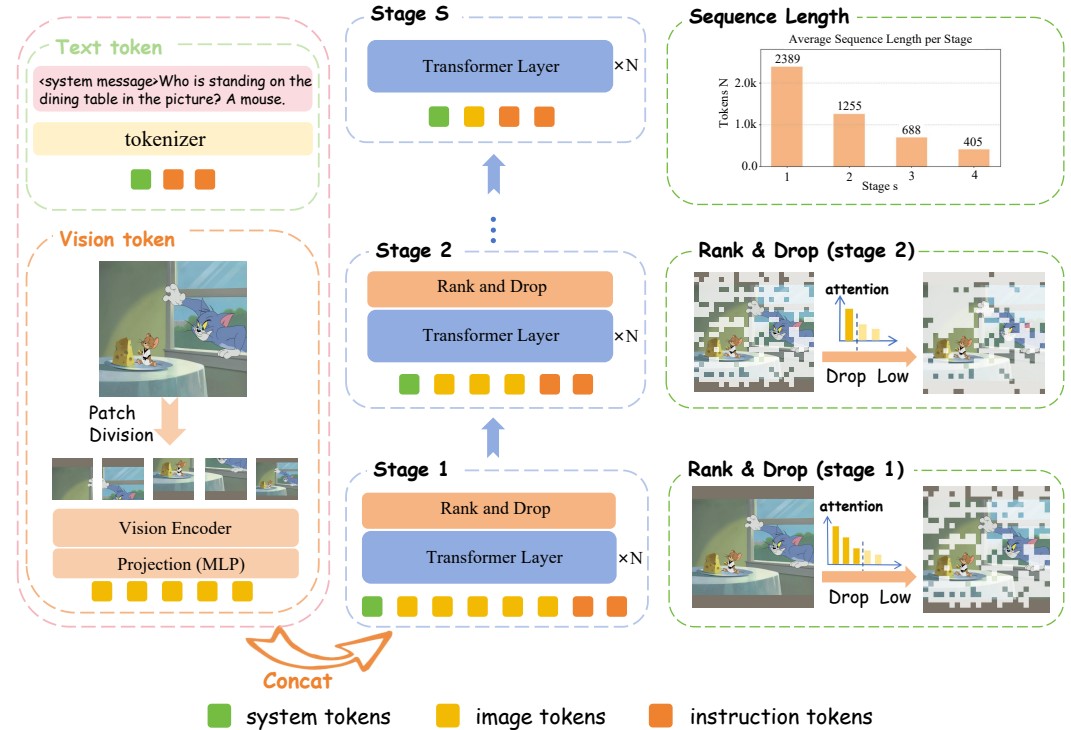

Figure 2: Overview of PyramidDrop. We divide the forward pass of the LLM into multiple stages, and drop part of the image tokens at the end of each stage with a pre-defined ratio. The dropping is based on a lightweight attention calculation with a negligible time overhead, and according to this criterion, the LLM accurately selects important image tokens related to instruction. Due to the efficient redundancy reduction strategy, the average sequence length decreases rapidly.

tokens, indicating that the model has already captured the necessary image information and the image tokens are redundant for the model now.

We further validate our hypothesis with an attention map comparison between different layers. As shown in Figure 1(b), the LVLM pays attention to most of the image tokens at shallow layers and the attention to different tokens shows a uniform pattern. On the contrary, at the middle of the LVLMs, the attention shows a sparse pattern and mainly focuses on the question related image local parts.

## 3.2 PYRAMIDDROP

Previous research on image token compression typically drops image tokens before passing them to the language model or uses a fixed compression ratio across all language model layers. However, as we analyzed in Sec 3.1, redundancy is not consistent across different layers. Redundancy of image tokens is relatively minimal in the shallow layers and becomes progressively larger in deeper layers. Thus, uniformly compressing image tokens across layers may lead to the loss of valuable information in the shallow layers while retaining unnecessary redundancy in the deeper layers.

Inspired by this observation, we propose PyramidDrop, which fully leverages layer-wise redundancy to compress image tokens. The pipeline of the proposed PyramidDrop is illustrated in Figure 2. To maximize training efficiency while preserving the essential information of the image tokens, we choose to divide the forward pass of the LLM into multiple stages. In the shallow layers, we retain a higher proportion of image tokens to preserve the entire vision information. At the end of each stage, we partially drop the image tokens, until nearly all the image tokens being eliminated in the deeper layers. This approach allows us to optimize training efficiency while maintaining critical information.

**LVLM Pre-fill Formulation.** We denote the vision encoder as $\mathcal{V}$, the vision-language projector as $\mathcal{P}$, the language model as $\mathcal{L}$, a pretrained LVLM as $\mathcal{M} = (\mathcal{L}, \mathcal{V}, \mathcal{P})$, where $\mathcal{L} = (\mathcal{L}_0, \mathcal{F})$. The language model consists of tokenizer $\mathcal{L}_0$ and $J$-layer transformer decoder $\mathcal{F}$. We formulate an image-text pair as $(\mathcal{V}, \mathcal{T})$, where the text is composed with an instruction and an answer $\mathcal{T} = \{T_i; T_a\}$[1]. The input of the transformer $\mathcal{F}$ contains both the image tokens $v_0 = \mathcal{P}(\mathcal{V}(v))$ and the text tokens $t_0 = \mathcal{L}_0(T)$.

During the forward pass of tokens, we can obtain the hidden states $v_j$, $t_j$ of vision tokens and text tokens in layer $j$, formally:

$$v_j, t_j = \mathcal{F}_j(v_{j-1}, t_{j-1}) \tag{1}$$

**Pyramid Visual Redundancy Reduction.** We partition the language into $\mathcal{S} = \{s_n\}_{n=0}^{S}$ stages, and remove the image tokens $v$ with a pre-defined ratio $\lambda$ at the end of each stage. Formally, with the image tokens $v_{s_n}$ as the input of stage $s_n$, we remove $\lceil (1 - \lambda)|v_{s_n}| \rceil$ tokens from the $v_{s_n}$ and treat the rest image tokens as the next stage input $v_{s_{n+1}}$.

Following our observation in Sec 3.1, the attention value between image and text tokens could reflect the image token importance properly, so we based on it to realize the drop operation. With the concern of calculation efficiency and training-inference consistency, we calculate the attention between all the image tokens and the last token of the instruction (we denote it as $t_j^I$, the last-instruction token in the following).

Formally, we denote the last layer of stage $s_n$ as $F_j$, we obtain key states of the image tokens as $k_j^v$ and the query state of last instruction token $q_j^{t_I}$ with the following operation:

$$k_j^v = \mathcal{K}_j(v_j), \quad q_j^{t_I} = \mathcal{Q}_j(t_j^I). \tag{2}$$

where $\mathcal{Q}_j$, $\mathcal{K}_j$ are the query matrix and the key matrix reused from the self-attention block of $F_j$.

We calculate the similarity with $q_j^{t_I} \times (k_j^v)^T$ and drop part of the image tokens based on the drop ratio $\lambda$. The image token number decreases exponentially stage by stage, and close to zero in the deeper layers. We denote the image token number of $v_0$ as $V = |v_0|$, and the image token number at each stage $V_s$ could be calculated as:

$$V_s = V_0 \cdot \lambda^{s-1}, \quad s = 1, 2, \ldots, S$$

**Efficiency Analysis of PyramidDrop** Here we analyze the efficiency from two parts: the computation overhead introduced by PyramidDrop, and the input sequence computation cost economized by PyramidDrop.

The extra computation cost introduced by PyramidDrop mainly lay in the similarity computing for image token ranking. Benefiting from our design, the calculation is only between a query toke and $V_s$ image tokens, so its computation complexity is $O(n)$ and only $S-1$ times in the forward process. Further, we notice the importance of FalshAttention in practice, so we keep using it during training and extract the query and key token from the original forward to calculate our lightweight similarity matrix.

When it comes to the computation cost economized by PyramidDrop. With the consideration of FlashAttn (Dao et al., 2022), we roughly define the forward inference cost of a layer with $N$ image tokens as a linear function with a constant factor $c$ that $c \cdot L$, so the overall computation cost of an LVLM with $L$ layers is $c \cdot N \cdot L$. When using PyramidDrop with S stages and the ratio $\lambda$, the overall computation cost is:

$$\frac{1 - \lambda^S}{S \cdot (1 - \lambda)} \cdot c \cdot N \cdot L \tag{3}$$

For example, if $\lambda = 0.5$ and we reduce the redundancy with 4 stages, it could save nearly $53.2\%$ computation cost theoretically, and we find this setting has a neglectable performance influence for models in practice.

---

[1]Here we omit the system prompt and chat format for illustrative purposes

# 4 EXPERIMENT

## 4.1 SETUP

**Models**  We verify the effectiveness and generalize of the proposed PyramidDorp by experiment on LVLMs with different architectures and input resolution. In detail, we study LLaVA-1.5-Vicuna-7B (Liu et al., 2024c), LLaVA-NeXT-Vicuna-7B (Liu et al., 2024b). LLaVA-1.5 is the most widely used open-source LVLM backbone for research, which is designed with a simple yet effective architecture that maps the 576 image features from the CLIP encoder as the LLM input with a projector. LLaVA-Next is the high-resolution extension of LLaVA-1.5, which supports at most 2880 image tokens and has better high-resolution capability.

**Benchmarks**  To thoroughly evaluate our image token compression strategy, we conduct experiments across 14 benchmarks. The MME Benchmark (Fu et al., 2023) assesses the perception and cognitive abilities of LMMs. MMBench and MMBench-CN (Liu et al., 2023) are benchmarks that manually craft questions to evaluate vision-related reasoning and perception in both English and Chinese, respectively. SEED (Li et al., 2023b), generated with the aid of GPT-4, comprises a dataset of approximately 19,000 questions pertaining to images and videos. MM-Vet (Yu et al., 2023) leverages GPT-4 for a six-dimensional evaluation of LMM capabilities. In the realm of traditional VQA benchmarks, such as VQA-v2 (Goyal et al., 2017) and VizWiz (Gurari et al., 2018), are also utilized. Additionally, several benchmarks featuring higher-resolution visual content, including DocVQA (Mathew et al., 2021), ChartQA (Masry et al., 2022), InfographicVQA (Mathew et al., 2022), and TextVQA (Singh et al., 2019). Finally, MMStar (Chen et al., 2024b) presents tasks with strong visual dependency, minimal data leakage, and requires sophisticated multimodal capabilities.

**Efficientness Evaluation**  We consider both the training time efficiency evaluation and inference time throughout. For training efficiency, we report the real training GPU hours with the same devices. For inference throughout, we follow the FastV(Chen et al., 2024a) and report the FLOPs of the image token part. In detail, we consider the FLOPs of the multi-head attention and the feed-forward network modules as $4nd^2 + 2n^2d + 2ndm$, where $n$ is the number of tokens, $d$ is the hidden state size, and $m$ is the intermediate size of the FFN. Considering there are three linear layers in FFN of LLaMA, the FLOPs is modified as $4nd^2 + 2n^2d + 3ndm$. Our PyramidDrop has different image token numbers at different stages and the FLOPS could be calculated by:

$$\sum_{s=0}^{S-1} K_s \times \left(4n_s d^2 + 2n_s^2 d + 3n_s dm\right) \quad \text{s.t.} \quad n_s = \lambda^s \times n, \quad s = 0, 1, 2, \dots, S-1 \quad (4)$$

**Implementation details**  Given that the LLM within the LVLM used in our experiments consists of 32 layers, we employ a straightforward approach by fixing $S$ to 4, effectively dividing the LLM into four equal parts. This segmentation allows the forward pass to be divided into four stages, with the number of image tokens decreasing exponentially at each stage. During accelerated training, we can adjust the value of $\lambda$ to control the proportion of image tokens that are pruned, and by default, $\lambda = 0.5$. We conduct all the experiments on 8 NVIDIA A100 80GB GPUs.

It is important to note that, since the LLaVA-NeXT model's data and training code are not open-source, we conduct training based on the open-source project Open-LLaVA-NeXT (Lin & Long, 2024). Due to differences in a portion of the training data, the benchmark performance may vary compared to that of LLaVA-NeXT (Liu et al., 2024b) blog.

## 4.2 EFFICIENT OF PYRAMIDDROP IN TRAINING

**PyramidDrop is effective for diverse architectures.**  We first study the PyramidDrop on both LLaVA-1.5 and LLaVA-Next. As shown in Table 1, PyramidDrop reduces the training time (including both pretraining and fine-tuning stages) of the LLaVA-Next from 366 to 218 GPU hours, resulting in an impressive 40% reduction in overall time. Besides the promising efficiency improvement, the model's performance remains comparable to the original on 14 different benchmarks. Notably, for fine-grained benchmarks like TextVQA, DocVQA, and OCRVQA, images contain a large amount of text and even documents, which request a dense and fine-grained understanding of

Table 1: LVLM w and w/o our method on 6 benchmarks. Benchmark names are abbreviated due to space limits. MMB: MMBenchmark (Liu et al., 2023); MMB$^{CN}$: MMBench-Chinese (Liu et al., 2023); SEED$^I$: SEED-Bench (Image) (Li et al., 2023b)

| Model | Train & Infer | GPU hours | #patches | Infer Flops(T) | MME | MMB | MMB$^{CN}$ | SEED$^I$ | MM Star | POPE | Avg |
|---|---|---|---|---|---|---|---|---|---|---|---|
| LLaVA -NeXT-7B | vanilla | 366 | 5 | 20.8 | 1534.1 | 68.7 | 60.5 | 71.1 | 41.1 | 86.1 | 67.4 |
| | PDrop | 218 | 5 | 9.46 | 1540.8 | 67.8 | 60.6 | 69.9 | 41.7 | 86.5 | 67.3 |
| | vanilla | 483 | 9 | 40.6 | 1544.7 | 67.4 | 60.0 | 69.5 | 40.0 | 86.3 | 66.7 |
| | PDrop | 269 | 9 | 18.1 | 1542.0 | 68.1 | 61.0 | 70.3 | 40.9 | 86.6 | 67.3 |
| LLaVA -1.5-7B | vanilla | 104 | 1 | 3.82 | 1510.7 | 64.3 | 58.3 | 66.1 | 33.2 | 85.9 | 63.9 |
| | PDrop | 79 | 1 | 1.78 | 1467.3 | 66.1 | 58.5 | 65.5 | 34.0 | 86.0 | 63.9 |

Table 2: LLaVA -NeXT-7B on other 8 benchmarks. We report more benchmarks which contain lots of fine-grained content to examine the performance. We denote PyramidDrop as PDrop.

| Model | Train & Infer | GPU hours | #patches | Doc VQA | Info VQA | Text VQA | Chart QA | OCR VQA | VQA V2 | Viz Wiz | GQA | Avg |
|---|---|---|---|---|---|---|---|---|---|---|---|---|---|
| LLaVA -NeXT-7B | vanilla | 366 | 5 | 70.0 | 33.3 | 67.2 | 64.0 | 63.7 | 81.7 | 59.6 | 64.2 | 63.0 |
| | PDrop | 218 | 5 | 69.0 | 31.7 | 67.7 | 63.0 | 63.1 | 81.5 | 61.0 | 63.9 | 62.6 |
| | vanilla | 483 | 9 | 74.3 | 36.2 | 67.6 | 63.0 | 63.8 | 81.6 | 58.0 | 63.5 | 63.5 |
| | PDrop | 269 | 9 | 75.0 | 37.4 | 68.4 | 64.3 | 63.5 | 81.7 | 60.6 | 64.1 | 64.4 |

the image. Even in this case, our approach still maintain performance at the original level. This indicates that our method successfully compresses redundant information while preserving the most critical image content.

In the case of LLaVA-1.5, which processes fewer image tokens per sample, the acceleration is not as pronounced as with LLaVA-NeXT. However, it still offers a nearly 20% improvement in speed with comparable performance. This underscores the potential of our method to enhance training efficiency across different model configurations.

**PyramidDrop enables larger resolution with constrained cost.** The PyramidDrop is proposed to reduce the redundancy within image tokens, and as we observed above, it enjoys higher speedup with the increase of the image/text token ratio. In this part, we explore its performance with higher image/text token ratio. In detail, LLaVA-NeXT is designed with a flexible image processing strategy in which an image is divided into a maximum of four local patches and a global patch, leading to at most 2880 image tokens. We denote it as LLaVA-NeXT-p5 and experiment on the LLaVA-NeXT-p9 by increasing the maximum local patches into 8 patches.

As shown in Table 2, with the increased image/text ratio, PyramidDrop reaches a higher speedup that only 269 GPU hours is used for training, which is only 55% of the vanilla LLaVA-Next-p9. Besides the superb speedup, the model trained with PyramidDrop achieves a slightly higher average performance across the 14 benchmarks. We argue too many image tokens with redundant information may confuse the LVLMs and hinder their performance, while our PyramidDrop efficiently reduce the image tokens number and helps the LVLM to focus on the critical information. Furthermore, it is worth noting that the training time is even 70% of the original LLaVA-Next-p5 but achieves better performance on diverse tasks, showcasing the superb efficiency and effectiveness of PyramidDrop.

**PyramidDrop training encourages LVLMs to understand images compactly.** Then we dive into the properties of the model trained with PyramidDrop and conduct experiments to investigate the changes in image token redundancy. Two models are employed for this exploration: the vanilla LLaVA-1.5 and the LLaVA-1.5 trained with our approach. As illustrated in Figure 3, we plot the TextVQA scores against the retained image tokens at layers 2, 8, 16, and 24, maintaining the same experimental settings as Sec 3.1. We find that the curve of models trained with PyramidDrop keeps higher than the vanilla one. The phenomenon suggests that, for a given proportion of retained image tokens, model trained with PtramimdDrop preserves more image information and achieves better performance. Alternatively, at equivalent performance levels, our method allows for a higher ratio of

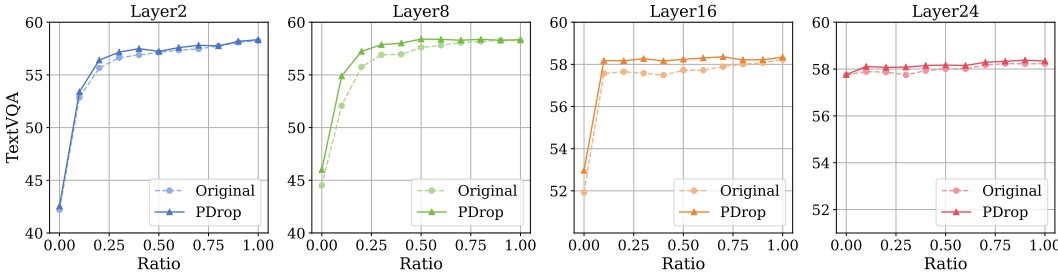

Figure 3: We compare the performance of the original LLaVA-1.5 and LLaVA-1.5 trained using PDrop, where we preserve different ratios of image tokens at layer 2, 8, 16, and 24, respectively. The horizontal axis represents the proportion of retained image tokens according to attention score.

Table 3: Performance gain with models trained with PyramidDrop. Directly applying efficient inference strategies like FastV to models trained with PyramidDrop yields substantial improvement.

| Model | Train | Infer | Infer Flops(T) | ChartQA | DocVQA | TextVQA | MME | SQA$^I$ | POPE | Average |
|---|---|---|---|---|---|---|---|---|---|---|
| | vanilla | vanilla | 20.8 | 64.0 | 70.0 | 67.2 | 1534.1 | 70.4 | 86.1 | 72.4 |
| LLaVA | PDrop | PDrop | 9.46 | 63.0 | 69.0 | 67.7 | 1540.8 | 70.1 | 86.5 | 72.2 |
| -NeXT-7B | vanilla | FastV | 10.6 | 55.9 | 62.1 | 66.0 | 1482.0 | 69.2 | 85.5 | 68.8 |
| | PDrop | FastV | 10.6 | 59.9 | 63.9 | 65.6 | 1492.7 | 68.9 | 86.8 | 70.0 |
| | Δ | | | **+4.0** | **+1.8** | -0.4 | **+0.5** | -0.3 | **+1.3** | **+1.2** |

Table 4: Ablation studies results. We adjust $\lambda$ form 0.4 to 0.6 for investigating the influence on performance and training time.

| Model | $\lambda$ | GPU hours | #patches | Infer Flops(T) | MME | MMB | GQA | MMB$^{CN}$ | SEED$^I$ | Doc VQA | Info VQA | Avg |
|---|---|---|---|---|---|---|---|---|---|---|---|---|
| | vanilla | 366 | 5 | 20.8 | 1534.1 | 68.7 | 64.2 | 60.5 | 71.1 | 70.0 | 33.3 | 63.5 |
| LLaVA | 0.4 | 204 | 5 | 8.22 | 1558.4 | 68.1 | 63.7 | 60.5 | 69.5 | 66.6 | 31.8 | 62.6 |
| -NeXT-7B | 0.5 | 218 | 5 | 9.46 | 1540.8 | 67.8 | 63.9 | 60.6 | 69.9 | 69.0 | 31.7 | 62.8 |
| | 0.6 | 240 | 5 | 11.0 | 1511.4 | 68.1 | 64.1 | 60.5 | 70.4 | 69.8 | 33.0 | 63.1 |
| | vanilla | 104 | 1 | 3.82 | 1510.7 | 64.3 | 62.0 | 58.3 | 66.1 | 21.4 | 20.4 | 52.6 |
| LLaVA | 0.4 | 75 | 1 | 1.54 | 1478.8 | 66.2 | 61.7 | 58.0 | 64.5 | 21.1 | 19.9 | 52.2 |
| -1.5-7B | 0.5 | 79 | 1 | 1.78 | 1467.3 | 66.1 | 61.9 | 58.5 | 65.5 | 21.5 | 20.2 | 52.4 |
| | 0.6 | 82 | 1 | 2.06 | 1471.8 | 65.9 | 62.0 | 58.9 | 65.1 | 22.5 | 21.0 | 52.7 |

image tokens to compress. This improvement can primarily be attributed to the multi-stage training strategy, which progressively prunes image tokens, encouraging the model to consolidate essential information into a smaller set of tokens, resulting in more densely informative representations.

We further validate our hypothesis by replacing the inference strategy with FastV. As demonstrated in Table 3, directly applying efficient inference strategies like FastV to models trained with PyramidDrop yields substantial improvements. Notably, there is a 1.3% increase in POPE and a 0.5% increase in MME, with even more pronounced gains observed on high-resolution benchmarks: ChartQA shows an increase of 4%, while DocVQA improves by 1.8%. These results provide compelling evidence for our hypothesis that training with PyramidDrop encourages the LVLMs to understand images compactly, which is a generalized result, rather than an overfit to the training strategy.

**Balancing PyramidDrop performance and efficiency with $\lambda$.** $\lambda$ balances the performance and efficiency of PyramidDrop, a larger $\lambda$ preserves more image information but slows down the training, and a smaller $\lambda$ has higher speedup while may influence the model performance. In this part, we study the influence of $\lambda$ on both LLaVA-1.5 and LLaVA-NeXT.

Table 5: Inference acceleration performance. We compare PDrop, FastV and vanilla model, and find PDrop outperforms FastV on almost all benchmarks. PDrop here is as an inference-only strategy.

| Model | Inference Strategy | TFLOPS | MME | SQA$^I$ | MMB$^{CN}$ | GQA | POPE | TextVQA | ChartQA | DocVQA | Avg |
|---|---|---|---|---|---|---|---|---|---|---|---|
| LLaVA -NeXT-7B | vanilla | 20.8 | 1534.1 | 70.4 | 60.5 | 64.2 | 86.1 | 67.2 | 64.0 | 70.0 | 69.9 |
| | FastV | 10.6 | 1482.0 | 69.2 | 60.0 | 63.0 | 85.5 | 66.0 | 55.9 | 62.1 | 67.0 |
| | PDrop | 9.5 | 1533.0 | 69.4 | 59.9 | 63.9 | 86.4 | 67.0 | 59.1 | 65.6 | 68.5 |
| | △ | | +2.5 | +0.2 | +0.1 | +0.9 | +0.9 | +1.0 | +3.2 | +3.5 | +1.5 |
| LLaVA -1.5-7B | vanilla | 3.82 | 1510.7 | 66.8 | 58.3 | 62 | 85.9 | 58.2 | 18.2 | 21.4 | 55.8 |
| | FastV | 2.01 | 1475.6 | 68.5 | 56.8 | 59.6 | 84.8 | 57.1 | 17.8 | 19.2 | 54.7 |
| | PDrop | 1.78 | 1500.8 | 69.2 | 58.5 | 60.1 | 84.8 | 57.5 | 18.6 | 21.1 | 55.6 |
| | △ | | +1.3 | +0.7 | +1.7 | +0.5 | +0.0 | +0.4 | +0.8 | +1.9 | +0.9 |

As shown in Table 4, we vary the $\lambda$ from 0.4 to 0.6 and report the model performance on both general and high-resolution benchmarks. For the general benchmarks, we observe a relative robust performance among different lambda, this indicates that for most questions, the information within images is somewhat redundant. When it comes to the DocVQA, which requires a fine-grained understanding on high-resolution images, the model performance shows a clear decline when the $\lambda$ decreases to 0.4. It is reasonable as the loss of critical image information and we could anticipate a more pronounced performance decline with the $\lambda$ keeps decreasing. Therefore, we opt for $\lambda = 0.5$, which maintains comparable performance to the baseline while also yielding a significant reduction in processing time.

### 4.3 EFFICIENT OF PYRAMIDDROP IN INFERENCE

**PyramidDrop outperforms SOTA methods as a inference-only strategy .** As illustrated in Table 5, we directly apply the multi-stage compression strategy during the inference phase of the vanilla model, comparing it with the inference acceleration approach, FastV. The results on LLaVA-Next demonstrate that our method significantly outperforms FastV across various critical benchmarks. Specifically, we achieve an impressive score of 1533.0 on MME, surpassing Fastv by 2.5%, while also exceeding it by 0.9% on both POPE and GQA. Notably, the advantages of our method become even more pronounced in high-resolution benchmarks. For instance, on the relatively challenging DocVQA, our approach outperforms FastV by 3.5%, and on ChartQA and TextVQA, we achieve improvements of 3.2% and 1% respectively.

Results from LLaVA-1.5 reveal similar trends across multiple benchmarks, including MME, ScienceQA, and MMBenchCN, where our method not only demonstrates superior performance but also achieves a greater reduction in FLOPs. When compared to the baseline, our approach consistently reaches comparable performance levels across most benchmarks, while effectively mitigating information loss in high-resolution benchmarks. These findings indicate that FastV's premature compression of image tokens leads to inevitably image information loss and significant performance declines in many benchmarks, whereas our multi-stage compression strategy preserves critical information from image tokens while maximizing the elimination of redundancy. The observation is also consistent with our finding in Sec 3.1 that in shallow layers, most image tokens are critical for LVLMs to understand the image properly, while in the deep layers, most of them are redundant for the LVLMs.

**PyramidDrop enjoys a better trade-off between performance and inference cost.** We further compare PyramidDrop and FastV under a precise FLOPs-constrained setting with LLaVA-NeXT-7B. In practice, we adjust the drop rate of FastV and the $\lambda$ of our PyramidDrop to control the model inference FLOPs and evaluate the model benchmark performance. As the FLOPs-performance curve shown in Figure 4, our PyramidDrop consistently outperforms FastV under different settings and across diverse benchmarks. For example, under a constraint of 12 TFLOPs, PyramidDrop outperforms FastV with 3.0% on DocVQA and 2.6% on ChartQA. When we reduce the inference cost to only 8 TFLOPs, the performance gap increases, with PyramidDrop surpassing FastV by 6% on DocVQA, and 5.9% on ChartQA. The results further prove that our multi-stage redundant reduction strategy matches the nature of LVLMs and enables the model to understand the image better under constrained inference cost.

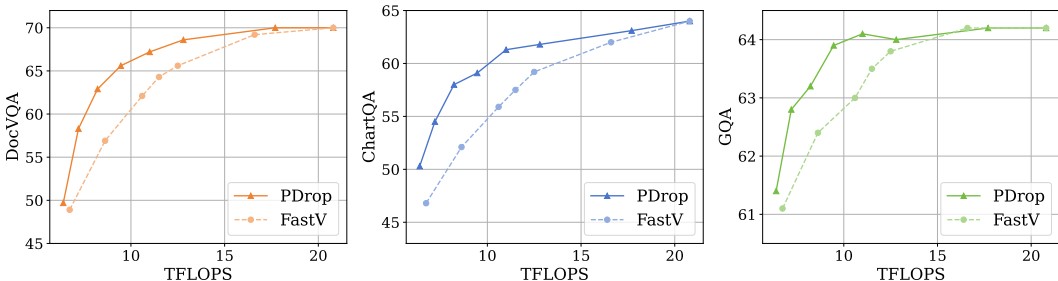

Figure 4: The performance of LLaVA-NeXT-7B with different inference acceleration strategies. PDrop (without training) outperforms FastV on DocVQA, ChartQA, and GQA with across various inference cost budgets.

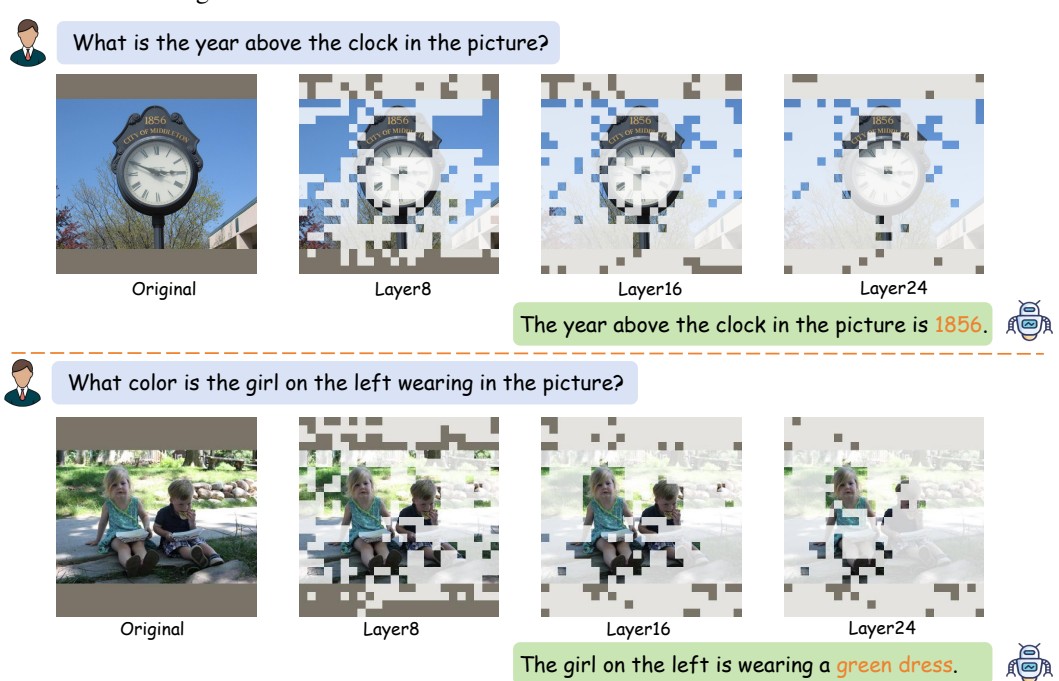

Figure 5: Visualization of token dropping in LLM of LLaVA -1.5. We compute the attention score of image tokens received from the last instruction token as the ranking criterion, and find LLM accurately retain image tokens according to instruction.

**LVLM with PyramidDrop effectively preserves image tokens related to instruction.**    As shown in Figure 5, we visualize the image tokens retained by LLaVA-1.5 with PyramidDrop in different stages. It is evident that when the user asks about a small object in the image, the LLM accurately identifies the region containing the relevant information based on the instructions and provides the correct answer. This demonstrates that our method effectively leverages the LLM's nature to understand images. The token dropping in PyramidDrop applied during inference does not result in the loss of valuable information.

## 5   CONCLUSION

We have introduced PyramidDrop, a simple yet effective strategy for reducing visual token redundancy in large vision-language models (LVLMs) to enhance efficiency with negligible performance loss. Our empirical study reveals that while all visual tokens are necessary in the shallow layers of LVLMs, token redundancy progressively increases in deeper layers. Extensive experiments demonstrate that PyramidDrop can achieve significant acceleration in both training and inference.

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
