# OpenReview forum: "PyramidDrop: Accelerating Your Large Vision-Language Models via Pyramid Visual Redundancy Reduction"
_ICLR.cc/2025/Conference — ICLR 2025 Conference Withdrawn Submission_

### Official Review · Reviewer_uEgW · 2024-10-21

**Soundness:** 3
**Presentation:** 3
**Contribution:** 3
**Rating:** 3
**Confidence:** 4

**Summary:**

Current approaches focus on reducing the number of visual tokens either before or within the early layers of LVLMs (Large Vision-Language Models). This paper considers the inconsistency in visual redundancy across different layers and proposes the PyramidDrop, which effectively reduces redundant visual tokens in LVLMs. Extensive evaluations on multiple benchmarks demonstrate that this method enhances efficiency with negligible performance loss.

**Strengths:**

1. This paper is coherent and logical, effectively demonstrating that visual redundancy varies across different layers.  Based on this observation, the authors propose a hierarchical sparsification strategy.
2. The experiments are comprehensive, with the proposed algorithm being validated across multiple benchmarks, demonstrating its feasibility.

**Weaknesses:**

1. The `method` section lacks clarity, particularly in explaining the rationale behind using the last instruction token and all visual tokens to calculate attention. Was there any attempt to use other indexes of the instruction tokens for attention calculation?
2. The approach introduces additional overhead, as the attention between the instruction token and visual tokens can be directly derived from the transformer's attention without the need for additional key and value computations.
3. The comparative experiments are incomplete, as only inference-only algorithms are compared, lacking a comparison with similar train-inference token reduction approaches, like LLaMA-VID [1] and VoCo-LLaMA [2].

[1] Li, Yanwei, Chengyao Wang, and Jiaya Jia. "Llama-vid: An image is worth 2 tokens in large language models." In *European Conference on Computer Vision*, pp. 323-340.

[2] Ye X, Gan Y, Huang X, Ge Y, Shan Y, Tang Y. VoCo-LLaMA: Towards Vision Compression with Large Language Models. arXiv preprint arXiv:2406.12275. 2024 Jun 18.

**Questions:**

1. How were the experimental results for the 0.0 ratio in Figure 1(a) obtained? Was the attention output of that layer directly used as the input for the LLM?
2. Since only the last instruction token is used to compute attention with all visual tokens, shouldn’t the visualization in Figure 5 tend to focus visual tokens on the last instruction token? Why does the attention highlight information such as "1856" and "green dress"?
3. Could you provide the practical speed comparison of the baseline and proposed method?

---

### Official Review · Reviewer_Gfov · 2024-10-23

**Soundness:** 3
**Presentation:** 2
**Contribution:** 2
**Rating:** 3
**Confidence:** 4

**Summary:**

The paper discusses the significant computational costs of processing images in large vision-language models (LVLMs) and proposes a new method called PyramidDrop, which reduces visual tokens across different layers to improve training and inference efficiency with minimal performance loss. PyramidDrop prunes image tokens progressively across model stages using a lightweight similarity calculation. Experiments show that it speeds up training and inference with results comparable to those of the original model.

**Strengths:**

1. The paper discusses the problem of computational efficiency in large vison-language models, which is a hot issue that urgently requires effective methods to accelerate speed.
2. The proposed method is easy to follow and replicate.

**Weaknesses:**

1. The contributions of this paper are not enough. The method is common, as it only uses rank and drop operations, like FastV, but with a hierarchical version. Additionally, the experimental results do not show particularly impressive outcomes, compared to other efficient methods that have been proven to work, such as VoCo-LLaMA[1], LLaMA-VID[2], or LLaVA-PruMerge[3].
2. Experiments are insufficient. In the training stage, this paper only compared the baseline with LVLM without rank and drop, with a lack of comparison with other efficient methods as point 1 demonstrated. In the ablation study, there is also no analysis of the impact of the number of layers chosen for token dropping.
3. There are some small typos, such as play a important in line 159, FalshAttention in line 256, and PtramimdDrop in line 377.

[1] Ye, Xubing, et al. "VoCo-LLaMA: Towards Vision Compression with Large Language Models." arXiv preprint arXiv:2406.12275 (2024).

[2] Li, Yanwei, Chengyao Wang, and Jiaya Jia. "Llama-vid: An image is worth 2 tokens in large language models." European Conference on Computer Vision. Springer, Cham, 2025.

[3] Shang, Yuzhang, et al. "Llava-prumerge: Adaptive token reduction for efficient large multimodal models." arXiv preprint arXiv:2403.15388 (2024).

**Questions:**

1. Experiments are insufficient. In the training stage, this paper only compared the baseline with LVLM without rank and drop, lack of comparison with other efficiency training methods, such as VoCo-LLaMA, LLaMA-Vid or LLaVa-PruMerge, etc.
2. In the ablation studies, there is also no analysis of the impact of the number of layers chosen for token dropping. In addition, have you tried different pruning ratios in different stages instead of using the same ratio in all stages?
3. I think PyramidDrop is a hierarchical token-pruning version of FastV. Can you describe more about the difference between your method and FastV?
4. Could you add the visualization of layer 2 in Figure 5?

---

### Official Review · Reviewer_H5EU · 2024-11-01

**Soundness:** 3
**Presentation:** 2
**Contribution:** 2
**Rating:** 3
**Confidence:** 5

**Summary:**

This paper introduces PyramidDrop to adaptively drop image tokens in multimodal models. They partition the layers into fixed groups and use a hyperparameter to indicate the ratio of tokens to keep during training and inference.

This paper is a natural extension of FastV with more fine-grained operations to prune the image tokens progressively.

This paper achieves better performance than FastV on all benchmarks. At the same time, this method is compatible with FastV.

**Strengths:**

1. The idea is simple and straightforward
2. The writing is easy to follow. The writing logic is clear.
3. This paper beats FastV at the inference stage.

**Weaknesses:**

1. The overhead of token pruning operation. Though this operation does not cost much computation,  more analysis on this is important since we want global speedup.

2. Speedup metric. Previous token pruning methods[1, 2] provide both GFLOPs and global latency. This work provides only GFLOPs which does not convince me of the actual speedup on hardware.

3. The effect of group size. As the main contribution of this paper, it does not include the discussion of the choice of group number and how will that influence performance and efficiency.

4. As shown in Table 5, the performance improvement is limited(<1.0) compared with FastV while FastV does not require multiply-time token pruning.

Overall this work is simple, but I am concerned that it may not meet the bar of ICLR.

Minor:

line 255: FalshAttention

line 253: toke

line 377: PtramimdDrop

[1] Bolya, Daniel, et al. "Token merging: Your vit but faster." arXiv preprint arXiv:2210.09461 (2022).

[2] Kong Z, Dong P, Ma X, et al. Spvit: Enabling faster vision transformers via latency-aware soft token pruning[C]//European conference on computer vision. Cham: Springer Nature Switzerland, 2022: 620-640.

**Questions:**

See above.

---

### Official Review · Reviewer_YQm5 · 2024-11-03

**Soundness:** 2
**Presentation:** 3
**Contribution:** 2
**Rating:** 3
**Confidence:** 5

**Summary:**

This paper introduces PyramidDrop, a method to improve the efficiency of Large Vision-Language Models by progressively reducing visual tokens across model layers. Specifically, PyramidDrop addresses computational efficiency in two ways: 1) it divides the LVLM into several stages and preserves all visual tokens in shallow layers, 2) it then progressively drops tokens at the end of each stage based on a lightweight attention-based similarity calculation. The authors demonstrate that visual token redundancy increases in deeper layers of LVLMs, making this staged approach more effective. Experiments on LLaVA-NeXT show that PyramidDrop reduces training time by 40% while maintaining comparable performance across 14 different benchmarks. The method is also shown to be effective for inference acceleration, reducing FLOPs by 55% and enabling training with doubled input resolution while using only 70% of the original training time, with better performance on high-resolution tasks.

**Strengths:**

1. The proposed method improve LVLM efficiency through progressive token reduction based on layer-wise redundancy analysis. Different from compressing tokens uniformly or only in early layers, PyramidDrop's stage-wise reduction aligns with the redundency distribution of visual tokens in LVLMs.

2. Experiments show the proposed method effectively reduce LLaVA-NeXT training time by 40% while maintaining performance across 14 benchmarks. It also enables training with doubled input resolution using only 70% of the original computational cost, demonstrating better efficiency-performance trade-off than existing methods like FastV.

3. The proposed method could be deployed as a plug-and-play module without requiring architectural modifications or additional training.

4. The method's design is simple yet effective, using only lightweight attention calculations for measuring token importance.

**Weaknesses:**

1. The technical novelty is very limited. The core idea of progressive token pruning has been previously explored in both LTP (Learned Token Pruning) [1] and Magic Pyramid[2] papers. LTP introduced learnable thresholds for token pruning across different layers, while Magic Pyramid combined progressive token pruning with early exiting. PyramidDrop's approach of stage-wise token reduction follows a similar progressive pruning strategy, making its technical contribution incremental.

2. Another critical limitation of the proposed method is its incompatibility with FlashAttention, as it relies on explicit attention score for token dropping. Without analyzing this limitation or providing comparisons against FlashAttention-powered baselines, the paper leaves open questions about the method's true efficiency gains in terms of memory usage and inference latency.

**Questions:**

See weakness for details.

---

### Note · Authors · 2024-11-14

I have read and agree with the venue's withdrawal policy on behalf of myself and my co-authors.